# The psychological burden of NMOSD – a mixed method study of patients and caregivers

Darcy C. Esiason[1], Nicole Ciesinski[2], Chelsi N. Nurse[2], Wendy Erler[3], Tom Hattrich[3], Ankita Deshpande[3], C. Virginia O'Hayer[1]*

1 Esiason O'Hayer Institute for Behavioral Medicine, Philadelphia, Pennsylvania, United States of America, 2 Department of Psychiatry and Human Behavior, Thomas Jefferson University Hospital, Philadelphia, Pennsylvania, United States of America, 3 Alexion Pharmaceuticals, Boston, Massachusetts, United States of America

* virginia.ohayer@jefferson.edu

**Data Availability Statement:** All relevant data are within the manuscript and its Supporting Information files.

## Abstract

Neuromyelitis optica spectrum disorder (NMOSD) is an inflammatory disorder of the central nervous system with common symptoms of rapid onset of eye pain, loss of vision, neck/back pain, paralysis, bowel and bladder dysfunction and heat sensitivity. The rare, unpredictable, and debilitating nature of NMOSD constitutes a unique psychological burden for patients and their caregivers, the specific nature and extent of which is not yet known. This mixed methods study, informed by both quantitative and qualitative data collected via self-report measures, focus groups, and in-depth interviews, aims to investigate and understand the psychological burden of patients with NMOSD and their caregiver/loved ones, so as to inform a specialized intervention. 31 adults living with NMOSD and 22 caregivers of people with NMOSD in the United States and Canada, recruited from NMOSD patient advocacy groups, social media groups, and through word of mouth from other participants, completed a battery of standardized self-report measures of anxiety, depression, trauma, cognitive fusion, valued living, and coping styles. Semi-structured focus group sessions were conducted via HIPAA-compliant Zoom with 31 patients, and separate focus groups were conducted with 22 caregivers. A subset of these samples, comprised of 16 patients and 11 caregivers, participated in individual semi-structured interviews, prioritizing inclusion of diverse perspectives. Descriptive statistics and bivariate correlations were run on quantitative self-report data using SPSS [Version 28.0.1]; data were stored in REDCap. Reflexive thematic analysis was employed regarding qualitative individual interview data. The majority of patients reported experiencing anxiety, depression, cognitive fusion, over-controlled coping, and lack of values-based living. Caregivers also reported heightened anxiety, cognitive fusion, and over-controlled coping, although they did not endorse clinically significant depression. Patient and caregiver degree of anxiety and of overcontrolled coping were both strongly positively correlated, likely affecting how both parties manage NMOSD-related stressors, both individually and as a dyad. Patients reported more anxiety, depression, psychological inflexibility, and lack of values-based living, compared with caregivers. Patient and caregiver narrative themes included mistrust of medical professionals, lack of support

**Funding:** CVO'H, DCE, NKC, CNN received an award (awarded to CVO'H as the PI) from Alexion Pharmaceuticals to fund this study (alexion.com). While the funders had no role in study design, data collection & analysis, decision to publish, or preparation of the manuscript - they were given 30 days in which to review the manuscript prior to publication.

**Competing interests:** The authors have declared that no competing interests exist.

immediately following diagnosis, changes in relationships, deviation from values-based living, internalization of feelings, and avoidant coping strategies to manage the psychological burden of NMOSD. A novel mental health intervention targeting the specific psychological burden of life with NMOSD is proposed.

## Introduction

Neuromyelitis optica spectrum disorder (NMOSD), is an inflammatory disorder of the central nervous system, characterized by recurrent attacks of inflammation and damage in the optic nerves and spinal cord. Despite availability of newer diagnostic tests [1] and consensus regarding diagnostic criteria [2], misdiagnosis with Multiple Sclerosis (MS) and subsequent iatrogenic treatment that exacerbates an NMOSD attack, is often part of the diagnostic history of adults with NMOSD [3, 4]. NMOSD attacks are unpredictable in nature and cause rapid onset of eye pain or blindness, limb weakness/numbness/paralysis, pain, loss of bowel and bladder control, and prolonged nausea/vomiting/hiccups [5]. NMOSD is about 8 times more prevalent among women (9:1 ratio among individuals who are anti-AQP4 seropositive) and among individuals with African, Asian, Pacific Island, Polynesian or Caribbean genetic ancestry [6].

The rare, unpredictable, debilitating nature of NMOSD creates a unique psychological burden on patients and their loved ones, the specific nature and extent of which is largely unknown. While the empirical literature to date, a summary of which follows, consistently documents depression, anxiety, pain, and sleep concerns, there is not currently a sufficient depth of knowledge required in order to address the psychological burden of NMOSD via an effective mental health intervention. In addition to general knowledge about the presence of anxiety, for example, detailed understanding is needed regarding the nature, context, degree, and impact of anxiety on the lives of patients and their caregiver/loved ones. The present study is intended to deepen our understanding of the extent and nature of this phenomenon, including as experienced by caregiver/loved ones. It is important to determine which aspects of psychological burden impact patients, caregivers, or both, and to what extent, as well as what events and context associated with increased burden. Our ultimate goal is to inform and inspire an intervention to address the unique psychological burden of NMOSD.

Depression and anxiety are common among individuals with NMOSD, with 30% to 50% endorsing clinically significant depressive symptoms [7–10] and 33% endorsing anxiety [11]. NMOSD patients are twice as likely to experience Major Depressive Disorder and high suicidal risk, as compared with patients with Multiple Sclerosis [9]. Among individuals with NMOSD, elevated anxiety is predictive of poor sleep quality [12] and poor overall quality of life [12–15].

Pain, sexual dysfunction, and stigma substantially impact emotional well-being and quality of life among people with NMOSD [16]. Chronic pain affects over 80% of NMOSD patients [17], including in the absence of recent relapse [18]. Pain severity is the strongest negative predictor of quality of life, and the most common symptom reported by NMOSD patients [18]. Three-quarters of men and women with NMOSD report sexual dysfunction, specifically reduced libido, decreased orgasm, and erectile dysfunction [19, 20]. Further contributing to the psychological burden, 60% of patients reported being affected by NMOSD-related stigma. Embarrassment due to physical limitations, perceived exclusion, avoidance/ostracism, and blame for their illness were deemed the most impactful manifestations of stigma, correlating with poor quality of life [21].

The psychological impact of NMOSD on caregivers and loved ones remains under-studied to date. A fifth of loved ones of NMOSD patients were found to be experiencing mild to moderately severe depressive symptoms [22]. Partners described pressure to take on new roles both inside and outside of the home during NMOSD relapses, limiting hobbies and activities to prioritize the patient's health, and both male and female partners reported challenges related to gender role shift [23].

The psychological burden of NMOSD is frequently mentioned by patients and loved ones in online forums, personal blogs, sub Reddits and social media groups. The inclusion of patient and caregiver voices allows a rich picture of the lived experience of people living with NMOSD and their caregivers. Patients and loved ones often discuss anxiety—particularly regarding future relapses [24, 25], difficulty obtaining a correct diagnosis [26] including misdiagnosis with MS—resulting in medical intervention that served to exacerbate NMOSD [27, 28], physical disability, emotional and financial impact on relationships [29, 30], and dealing with an "invisible illness" [31, 32] as central components of life with NMOSD.

Empirical literature and patient media perspectives suggest a significant unique psychological burden associated with diagnosis of NMOSD, although the full nature and extent of this burden is unclear, including as it pertains to caregivers. There is a clear need to develop a comprehensive understanding of this experience, both from patient and caregiver perspectives, so as to develop a targeted coping skills intervention to effectively meet this burden. This mixed methods study, using a combination of quantitative psychometric test data and qualitative data derived from focus groups and in-depth interviews, aims to identify and categorize salient features of the psychological burden specific to NMOSD to best inform effective intervention approaches.

## Materials and methods

### Participants

All participants completed informed consent prior to enrollment in this mixed methods study. The study protocol, as well as all participant-facing materials, were approved by the Thomas Jefferson University Institutional Review Board prior to study administration. Thirty-one patient participants were recruited via private NMOSD patient advocacy groups (Sumaira Foundation, Connor B. Judge Foundation, Guthy-Jackson Foundation), patient social media groups, and through word of mouth from other participants. Twenty-two caregiver participants were primarily recruited by requesting participation from patient participants. Caregivers were defined as anyone serving in a close supportive role for the patient, and could include spouses, partners, adult children, siblings, parents, close friends, other close family members. Inclusion criteria for the study were: (1) 18 years or older, (2) living with a diagnosis of NMOSD or caring for a loved one with NMOSD, (3) able to understand English, and (4) Canadian or US citizen. There were no exclusion criteria. Patient and caregiver participants were compensated via $150 gift card for completion of self-report measures and participation in a focus group, consistent with NIH guidelines to provide reasonable compensation for participant's time, particularly in the context of rare diseases with limited populations [33–35], and comparative with recent focus groups and interviews in the context of NMOSD [36].

Individual interviews occurred concurrently with focus groups after the completion of the first focus group. For the interviews, purposive sampling was used to ensure representation of racial and ethnic diversity, in attempts to approximate a sample representative of the larger population of NMOSD patients. Focus group participants who identified as members of diverse racial/ethnic groups were invited to participate in individual interviews, to ensure inclusion and representation of their voices. Similarly, male participants were also invited to

participate in individual interviews. Both patient and caregiver participants were compensated via $150 gift card for completion of self-report measures and participation in an individual interview.

## Measures

Prior to participation in focus groups, patients and caregivers completed a demographics survey as well as psychometric measures of anxiety (BAI), depression (BDI-II), cognitive fusion (CFQ-13), valued living (VLQ), psychological flexibility (AAQ-II), trauma history (ACEs) and coping styles. Finally, participants completed items assessing openness to treatment.

**Anxiety.** Anxiety symptoms were assessed using the Beck Anxiety Inventory (BAI) [37], a 21-item self-report measure assessing symptoms of anxiety over the past week. Sample items include "nervous" and "frightened", with answer options ranging from 0 (not at all) to 3 (severely) and total scores ranging from 0–63, with a higher score indicating higher levels of anxiety symptoms. Scores of 8 or above suggest clinically relevant anxiety. The BAI demonstrated excellent internal consistency in the full sample ($\alpha = .92$).

**Depression.** Depressive symptoms were measured by the Beck Depression Inventory II [38]. This 21-item self-report measures depressive symptoms over the past 2 weeks. The BDI-II assesses affective, cognitive, behavioral, and somatic symptoms of depression and has good construct validity [38]. Item scores range from 0–3. Possible total scores range from 0–63, with higher scores indicating higher levels of depressive symptoms. Scores of 17 or above suggest clinical depression. The BDI-II demonstrated excellent internal consistency in the full sample ($\alpha = .90$).

**Cognitive fusion.** Cognitive fusion was measured by the Cognitive Fusion Questionnaire (CFQ-13) [39], a 13-item self-report assessment of the degree to which one is "hooked" or "fused" to one's thoughts. Responses range from 1 (not true at all) to 7 (always true). Possible scores range from 13–91, with higher scores indicating higher levels of cognitive fusion. Scores of 34 or above suggest clinically relevant cognitive fusion. The CFQ demonstrated good internal consistency in the present sample ($\alpha = .82$).

**Psychological flexibility.** Psychological flexibility was assessed using the Acceptance and Action Questionnaire (AAQ-II) [40], a 7-item measure of psychological inflexibility/experiential avoidance, with higher scores indicating greater inflexibility. Scores of 24 or above suggest clinically relevant psychological inflexibility. The AAQ demonstrated good internal consistency in the full sample ($\alpha = .89$).

**Valued living.** Valued living, a construct similar to purpose in life, measuring the degree to which one is living according to their chosen value was measured by the 10-item Valued Living Questionnaire [41]. The VLQ demonstrated acceptable internal consistency for rated importance ($\alpha = .73$) and good internal consistency for past-week practice of value ($\alpha = .81$).

**Trauma history.** History of trauma was measured by the Adverse Childhood Experience Questionnaire (ACEs) [42], a 10-item questionnaire assessing 10 categories of abuse, neglect, and household dysfunction, in childhood. Scores of 4 or above suggests clinically significant range of adverse childhood experiences. The ACE demonstrated acceptable internal consistency in the full sample ($\alpha = .76$).

**Coping style.** Coping style, in particular whether the respondent leans toward overcontrolled coping or undercontrolled coping, was measured by the Styles of Coping Word-Pairs [43], a 47-item forced-choice measure of overall personality style. It demonstrated good internal consistency in the present sample ($\alpha = .85$).

## Procedures

Focus groups were conducted via HIPAA-compliant Zoom teleconferencing platform, with video on. Focus groups lasted 60 minutes, were offered at convenient times (including evenings), and were facilitated by licensed clinical providers (CVO'H, DCE). Patient and caregiver focus groups were conducted separately, to ensure patient privacy, and so that caregivers felt free to engage in an honest discussion about challenging aspects of their role. Focus groups required a minimum of 3 and a maximum of 12 participants, to ensure adequate variety of responses and time for each member to participate.

Each focus group began with a vote via show of hands (or raised hand Zoom feature) regarding which topics were of interest and relevance to members. Topics were derived from review of empirical and patient voices literature as well as literature regarding mental health concerns among other chronic neurological conditions. Additional questions were included regarding the impact of the COVID-19 pandemic. Each group ended with the open-ended invitation to offer other information that facilitators may not have known to ask.

Individual interviews lasted from 45–60 minutes and were also conducted via HIPAA-compliant Zoom, by the same research team members, and were recorded for transcription. These semi-structured interviews were informed by themes and content emerging from the focus group component of this study, as well as by detailed review of both the empirical literature regarding psychosocial stressors associated with NMOSD and of patient-driven data as described earlier. One caregiver individual interview took place in written format only, as the caregiver was deaf and did not feel comfortable participating via phone or Zoom. In efforts to include her valuable perspective, she was offered participation via written response to the interview prompts.

## Data analysis

**Quantitative analyses.** SPSS [Version 28.0.1] statistical software was used to conduct all quantitative analyses. Data were stored in REDCap. Total scores and clinical cutoffs were calculated from the raw psychometric test data. Descriptive statistics were run on all demographic and psychological variables. All continuous variables were assessed for normality and linearity. Results of the Kolmogorov Smirnov and Shapiro-Wilk Tests demonstrated that the BAI and the coping style questionnaire total scores were non-normally distributed in the patient sample, whereas the coping style questionnaire and the ACE total scores were non-normally distributed in the caregiver sample. Thus, bootstrapping was implemented in all inferential analyses to adjust for non-normal distributions. Bivariate correlational analyses were run assessing the relationships among all psychological variables in the independent patient and caregiver samples. As the patient sample was 90% (n = 27) female, gender differences across psychological variables were probed only in the caregiver sample (52.4%, n = 11 female) with t-tests. Additionally, bivariate correlations and t-tests were run on paired patient-caregiver data to assess the relationships and differences between patients and their caregivers across the psychological variables.

**Qualitative analyses.** Semi-structured focus group sessions were conducted via HIPAA-compliant Zoom with 31 patients, and separate focus groups were conducted with 22 caregivers. A subset of these samples, comprising 16 patients and 11 caregivers, participated in individual semi-structured interviews, prioritizing inclusion of diverse perspectives.

Focus group data were analyzed through classical content analysis. A priori codes were developed prior to focus groups based on thorough review of both empirical and patient-based literature as described in the Introduction. During focus groups, an assistant moderator tallied the frequency of participants' use of a particular code. After focus groups, researchers used

audio files of focus groups to assign semantic data to each code. Focus group data informed the development of semi-structured interview guides for individual interviews. Distinct interview guides were developed for the patient interviews and for the caregiver interviews. Interviewers also allowed for open-ended discussion as guided by the patient and caregiver's prior answers and topics of interest. Individual interviews began after the first focus group concluded.

Individual interviews were recorded, transcribed, and analyzed according to the 6 steps of reflexive thematic analysis [44]: (1) familiarizing oneself with the data, (2) generating codes, (3) constructing themes, (4) reviewing potential themes, (5) defining and naming themes, and (6) producing the report [44]. One interview was conducted by writing only, as the spouse-caregiver is deaf.

Individual interviews were transcribed by two transcriptionists in the Thomas Jefferson University Department of Public Health. Investigators (DCE & CVO'H) read each transcript to familiarize with the data. Researchers developed two codebooks–one for patient interviews and one for caregiver interviews–with both deductive and inductive input, including codes derived from investigator clinical experience with rare disease psychology, review of empirical literature, codes derived from analysis of the focus groups, and codes generated from initial review of individual interviews. Using a roundtable approach, investigators (DCE & CVO'H) manually mapped themes as they were conceptualized from the data. Investigators (DCE, CVO'H, CNN, NKC and 3 graduate students) then refined the codebook and continued to review themes before a second roundtable was hosted to identify the overall narratives in the analysis.

## Results

### Psychometric test data—patients

Thirty out of 31 patient participants completed a demographics survey. One patient failed to turn in any questionnaire data. Twenty-two caregivers completed a caregiver version of the demographics survey. Summary statistics describing the focus group sample are listed in Table 1. See S2 Data.

Sixteen patients and eleven caregivers who participated in focus groups were invited to participate in individual interviews, prioritizing inclusion of diverse racial/ethnic and gender presentations. Sample summary statistics for individual interview participants are described in Table 2.

Descriptive statistics for all psychological variables among patients are summarized in Table 3. Patients endorsed a mean BDI-II score of 14 (SD = 9.5), suggesting mild mood disturbance. While 40% (n = 12) of respondents did not report depressive symptoms, 26.7% (n = 8) reported mild depressive symptoms, 20% (n = 6) endorsed borderline depressive symptoms, 6.7% (n = 2) reported moderate symptoms, and 6.7% (n = 2) reported severe depression (Fig 1). Overall, 33% (n = 10) of the sample endorsed symptoms suggestive of clinical depression.

Participants endorsed a mean BAI score of 14.2 (SD = 9.9), suggesting mild anxiety. While 27.5% (n = 8) reported minimal anxiety, 31% (n = 9) reported mild anxiety, 31% (n = 9) endorsed moderate anxiety, and 10.5% (n = 3) reported severe anxiety (Fig 2). Overall, 72% (n = 21) of the sample endorsed symptoms suggestive of clinically relevant anxiety.

The majority of participants endorsed normal levels of psychological flexibility on the AAQ-II (M = 16.6, SD = 7.5), with only 13.3% (n = 4) of the sample scoring in the range of clinically relevant inflexibility. However, most participants (63.3%, n = 19) scored in the clinical range for cognitive fusion on the CFQ-13 (M = 37.5, SD = 13.2), suggesting clinically relevant fusion (i.e., rigid attachment to thoughts as truth). Most participants reported using

**Table 1. Sociodemographic characteristics of participants in focus groups.**

| Baseline characteristics | Patients | | Caregivers | |
|---|---|---|---|---|
| | *n* | % | *n* | % |
| Gender | | | | |
| Female | 27 | 90 | 11 | 50 |
| Male | 3 | 10 | 11 | 50 |
| Racial/Ethnic Identity | | | | |
| Caucasian | 21 | 70 | 14 | 64 |
| Hispanic | 4 | 13 | 1 | 5 |
| Asian American/Pacific Islander | 3 | 10 | 1 | 5 |
| African American/Black | 2 | 6.7 | 3 | 14 |
| Other | 0 | 0 | 1 | 5 |
| Chose Not to Answer | 0 | 0 | 1 | 5 |
| AQP4 Positive | 80 | 24 | n/a | n/a |
| Hospitalization due to NMOSD* | | | | |
| Lifetime | 24 | 80 | 19 | 85 |
| Past Year | 7 | 23 | 5 | 20 |
| Past 1–3 Years | 0 | 0 | 2 | 10 |
| Past 3–5 Years | 8 | 29 | 8 | 35 |
| Over 5 Years Ago | 9 | 26 | 3 | 22 |
| Caregiver Relationship | | | | |
| Spouse/Partner | n/a | n/a | 8 | 36 |
| Parent | n/a | n/a | 5 | 23 |
| Adult Children | n/a | n/a | 4 | 18 |
| Sibling | n/a | n/a | 3 | 14 |
| Cousin | n/a | n/a | 1 | 5 |

*N* = 30 for patients, 22 for caregivers.

*Reflects the number and percentage of loved ones of caregivers who reported hospitalization due to NMOSD.

overcontrolled coping (82.7%, n = 24) on the Styles of Coping Word Pairs, with only 17.3% (n = 5) using undercontrolled coping.

Participants endorsed a variety of values on the VLQ, and reported living according to the following values they rated as important: parenting, work, education, spirituality and community. However, they also endorsed *not* living according to values which they rated as important, including physical health, recreation/fun, friends/social, marriage, and family.

A minority of patient participants (30%, n = 9) reported a clinically significant history of adverse childhood events on the ACEs (M = 2.4, SD = 2.4, normal range). Adverse childhood experiences endorsed by this sample include: parental divorce (50%, n = 15); abuse: verbal/emotional (33.3%, n = 10), physical (26.7%, n = 8), sexual (20%, n = 6); neglect: emotional (23.3%, n = 7), physical (6.7%, n = 2); household member with: substance abuse (30%, n = 9) mental illness (26.7%, n = 8) imprisoned (16.7%, n = 5); and caregiver physical abuse (6.7%, n = 2).

Bivariate correlational analyses revealed that patients endorsing greater psychological inflexibility also reported significantly higher levels of cognitive fusion ($r(28) = .690, p < .001$) as well as symptoms of anxiety ($r(28) = .553, p = .002$) and depression ($r(28) = .577, p < .001$). Similarly, greater cognitive fusion was significantly associated with increased symptoms of anxiety ($r(28) = .510, p = .052$) and depression ($r(28) = .506, p = .005$); however, it was also

**Table 2. Sociodemographic characteristics of participants in individual interviews.**

| Baseline characteristics | Patient | | Caregivers | |
|---|---|---|---|---|
| | *n* | % | *n* | % |
| Gender | | | | |
| Female | 13 | 81 | 5* | 45 |
| Male | 3 | 19 | 6 | 55 |
| Racial/Ethnic Identity | | | | |
| Caucasian | 10 | 63 | 8 | 73 |
| Hispanic | 1 | 6 | 1 | 9 |
| Asian American/Pacific Islander | 2 | 13 | 1 | 9 |
| African American/Black | 3 | 19 | 1 | 9 |
| Other | 0 | 0 | 0 | 0 |
| Chose Not to Answer | 0 | 0 | 0 | 0 |
| Caregiver Relationship | | | | |
| Spouse/Partner | n/a | n/a | 6* | 54 |
| Parent | n/a | n/a | 3 | 27 |
| Adult Children | n/a | n/a | 0 | 0 |
| Sibling | n/a | n/a | 2 | 18 |
| Cousin | n/a | n/a | 0 | 0 |

*N* = 16 for patients, 11 for caregivers.

*One spouse caregiver conducted individual interviews in writing only, as she is deaf.

**Table 3. Descriptive analysis of psychological variables in patients and caregiver samples.**

| Patients | | | | |
|---|---|---|---|---|
| | Mean | SD | Skewness | Kurtosis |
| Depression (BDI-II) | 14.03 | 9.50 | 1.35 | 2.23 |
| Anxiety (BAI) | 14.24 | 9.92 | 0.88 | 0.60 |
| Psychological Inflexibility (AAQ-II) | 16.60 | 7.49 | 1.12 | 1.53 |
| Cognitive Fusion (CFQ-13) | 37.50 | 13.17 | 0.10 | -0.18 |
| Coping Style | — | — | — | — |
| Undercontrolled | 14.17 | 8.88 | 0.44 | -0.77 |
| Overcontrolled | 29.63 | 10.71 | -0.94 | 0.45 |
| Childhood Trauma (ACEs) | 2.40 | 2.39 | 1.05 | 0.82 |
| **Caregivers** | | | | |
| | Mean | SD | Skewness | Kurtosis |
| Depression (BDI-II) | 7.20 | 5.92 | 0.46 | -0.66 |
| Anxiety (BAI) | 8.10 | 9.31 | 1.08 | 0.03 |
| Psychological Inflexibility (AAQ-II) | 11.90 | 4.84 | 0.85 | 0.17 |
| Cognitive Fusion (CFQ-13) | 32.10 | 12.19 | -0.22 | -1.18 |
| Coping Style | — | — | — | — |
| Undercontrolled | 10.71 | 4.99 | 1.08 | 0.23 |
| Overcontrolled | 32.19 | 9.27 | -1.26 | 0.77 |
| Childhood Trauma (ACEs) | 2.00 | 2.03 | 0.59 | -1.05 |

*N* = 30 for patients, 21 for caregivers.

SD = standard deviation

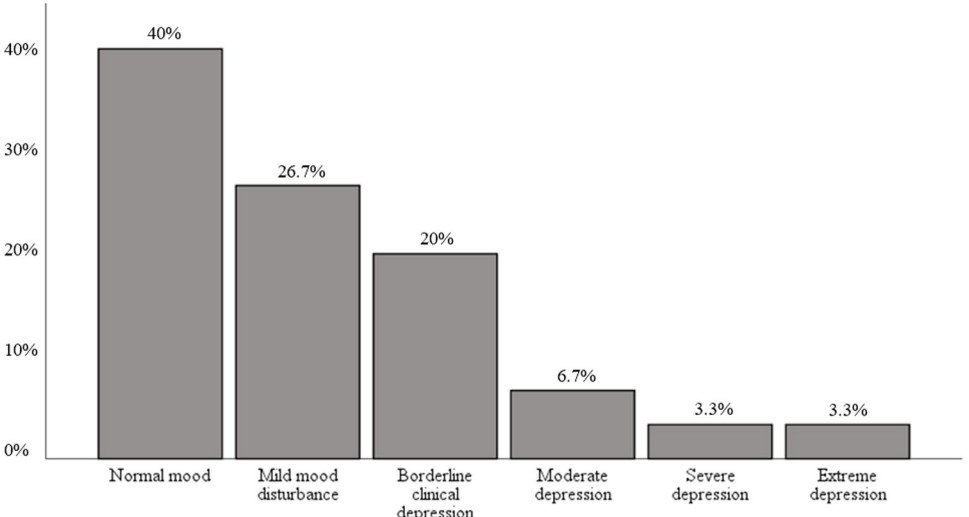

**Fig 1. NMOSD patient depressive symptoms endorsed on BDI-II.**

associated with lower endorsement of an overcontrolled coping style ($r(28)$ = -.388, $p$ = .038). Finally, patients who reported higher levels of anxiety also reported higher levels of depression ($r(28)$ = .555, $p$ = .002). Extent of childhood adverse experiences was not found to be significantly associated with scores on measures of anxiety, depression, psychological inflexibility, cognitive fusion, or coping style.

Most patients (62.2%, n = 23) reported having engaged in talk therapy in the past. However, only 26.9% (n = 7) found this "very helpful" in meeting their needs, with 57.5% (n = 15) finding it only "somewhat helpful" and 15.4% (n = 4) finding it "not helpful at all". The majority of patients expressed interest in an NMOSD-specific telehealth-delivered talk therapy, with 67.6% (n = 25) expressing definite interest, and 27% (n = 10) indicating they might be

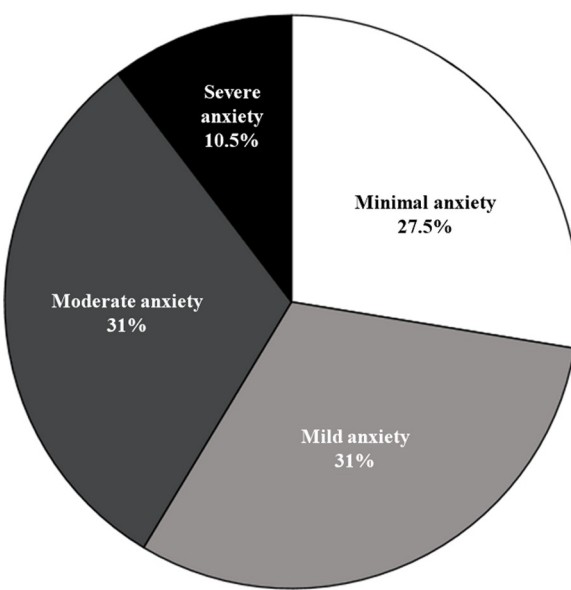

**Fig 2. NMOSD patient anxiety symptoms endorsed on BAI.**

interested. Only 5.4% (n = 2) were not interested. When asked if they would like to include their caregiver/loved one in such a treatment, most patients wanted their caregiver involved in some capacity, whether attending some therapy sessions (32.4%, n = 12), all sessions (13.5%, n = 5), or having the caregiver attend separate sessions (21.7%, n = 8). A minority of patients (8.9%, n = 7) did not want their caregiver to attend, or did not have a caregiver (13.5%, n = 5).

## Psychometric test data–caregivers

Twenty-one out of 22 caregiver participants completed a demographics survey. One caregiver failed to complete self-report measures. Descriptive statistics for all psychological variables among caregivers are summarized in Table 3. See S1 Data. Caregivers endorsed a mean BDI-II score of 7.2 (SD = 5.9), suggesting no depressive symptoms. While 70% (n = 14) of respondents did not report any depressive symptoms, 25% (n = 5) endorsed mild mood disturbance, and 5% (n = 1) reported borderline depressive symptoms (Fig 3).

Caregivers endorsed a mean BAI score of 8.1 (SD = 9.3), suggesting mild anxiety. While 55% (n = 11) reported minimal anxiety, 25% (n = 5) reported mild anxiety, 15% (n = 3) endorsed moderate anxiety, and 5% (n = 1) reported severe anxiety (Fig 4). Overall, 45% (n = 9) of the sample endorsed symptoms suggestive of clinically relevant anxiety.

The vast majority of participants endorsed normal levels of psychological flexibility on the AAQ-II (M = 11.9, SD = 4.8), with only 5% (n = 1) of the sample scoring in the range of clinically relevant inflexibility. Half (n = 10) of all participants scored in the clinical range for cognitive fusion on the CFQ-13 (M = 32.1, SD = 12.2), suggesting clinically relevant fusion. Every caregiver reported using overcontrolled coping (100%, n = 20) on the Styles of Coping Word Pairs.

In the VLQ, caregivers reported living according to the following values they deemed important: physical health, work, family, and community. However, they endorsed *not* living according to the following values they rated as important: parenting, marriage, and spirituality.

A minority of caregivers (30%, n = 6) reported a clinically significant history of adverse childhood events on the ACEs (M = 2.0, SD = 2.0, normal range). Adverse childhood experiences endorsed by this sample include: parental divorce (28.6%, n = 6); abuse: verbal/

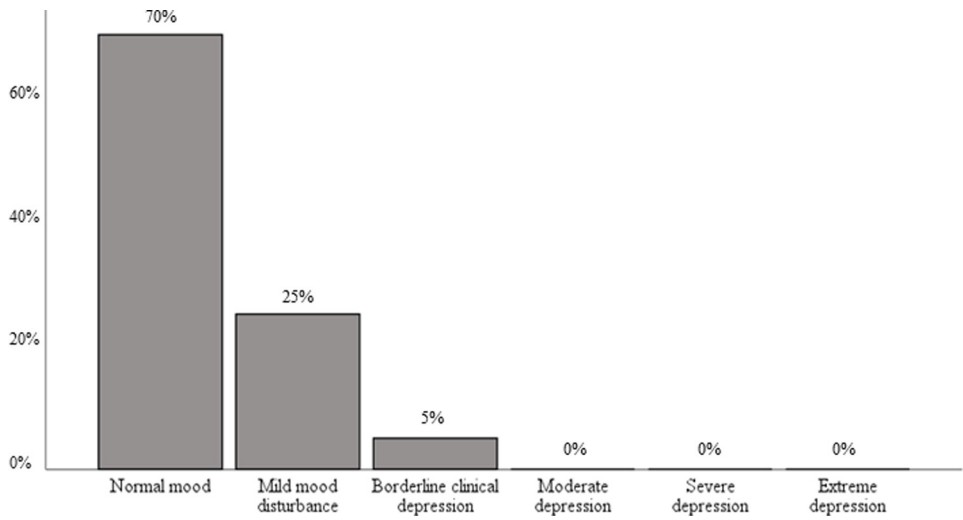

**Fig 3. Caregiver depressive symptoms endorsed on BDI-II.**

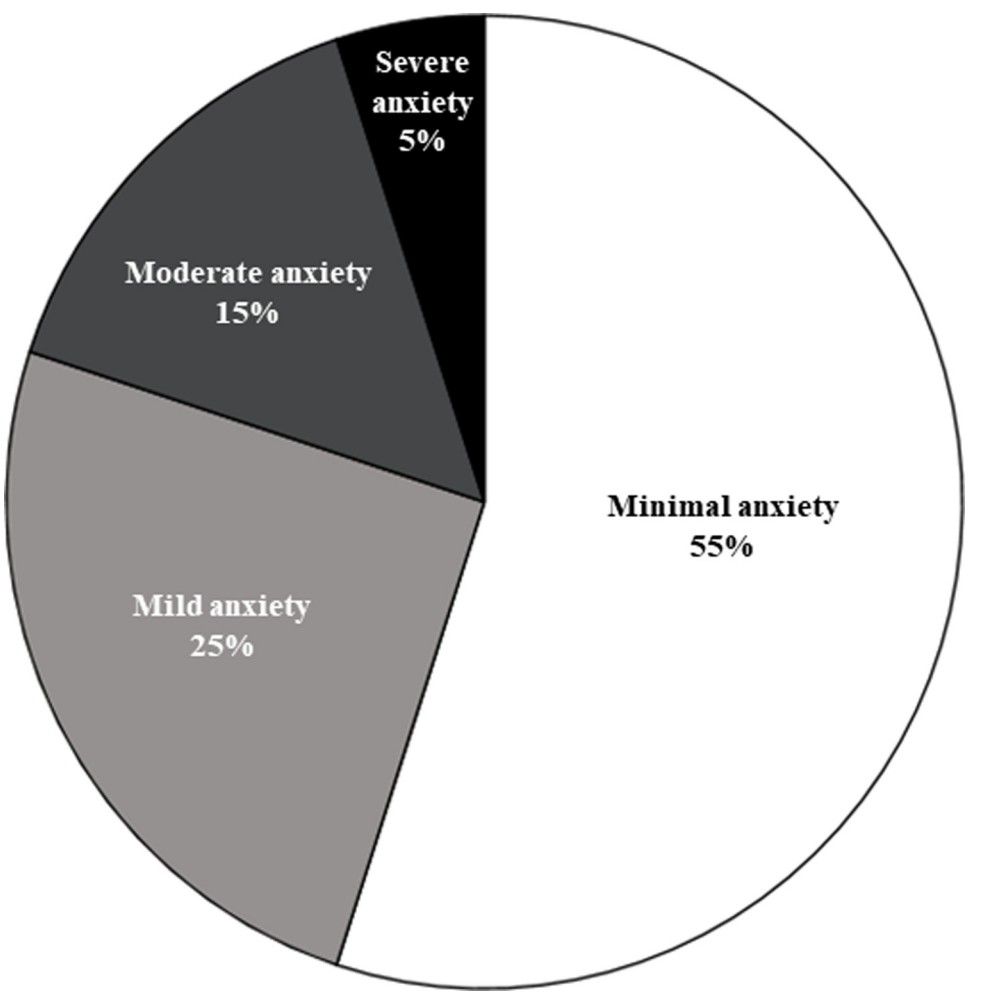

**Fig 4. Caregiver anxiety symptoms endorsed on BAI.**

emotional (38.1%, n = 8), physical (23.8%, n = 5), sexual (0%); neglect: emotional (23.8%, n = 5), physical (0%); household member with: substance abuse (42.9%, n = 9) mental illness (23.8%, n = 5) imprisoned (9.5%, n = 2); and caregiver physical abuse (9.5%, n = 2).

Bivariate correlations revealed that caregivers who endorsed greater cognitive fusion tended to also report heightened anxiety ($r(19) = .450$, $p = .046$) and an overcontrolled coping style ($r(19) = .490$, $p = .028$). Also, mirroring patient findings, higher levels of anxiety symptoms were significantly associated with higher endorsement of depressive symptoms ($r(19) = .502$, $p = .024$). No other significant associations were found among the caregiver self-report measures.

A significant minority caregivers (47.6%, n = 10) reported having engaged in talk therapy in the past. However, only 3 caregivers found prior therapy "very helpful" in meeting their needs, with 3 finding it only "somewhat helpful" and 2 finding it "not helpful at all". The majority of caregivers expressed interest in an NMOSD-specific telehealth-delivered talk therapy, with 40% (n = 8) definitely being interested, and 45% (n = 9) indicating they might be interested. Only 15% (n = 3) were not interested. When asked if they would like to include their loved one with NMOSD in such a treatment, most caregivers wanted their patient loved one to attend in some capacity, whether attending some therapy sessions (50%, n = 10), all sessions (30%, n = 6), or having the patient attend separate sessions (10%, n = 2). A minority of

caregivers (10%, n = 2) did not want their loved one with NMOSD to attend sessions with them.

No gender differences emerged across any variables in the caregiver sample.

## Paired patient-caregiver analyses

Bivariate correlations are summarized in Table 4. Higher patient anxiety was associated with higher caregiver anxiety, and higher level of overcontrolled coping style in patients was associated with higher level of overcontrol in caregivers. Additionally, patient depression scores were associated with lower levels of undercontrol in caregivers, whereas level of undercontrolled coping in patients was associated with higher psychological inflexibility in caregivers.

No other bivariate correlations emerged as significant.

Paired-samples t-tests demonstrated that patients endorsed significantly higher levels of psychological inflexibility ($t(18) = 2.111$, p = .049), anxiety ($t(18) = 3.465$, p = .003), and depression ($t(18) = 3.229$, p = .005) than their caregivers. No significant differences emerged between patients and their caregivers on cognitive fusion, coping style, and adverse childhood experiences. Additionally, patients endorsed larger discrepancies in valued living than their caregivers ($t(18) = 2.741$, $p = .05$).

## Qualitative data–themes from focus groups and interviews

A variety of themes were conceptualized across patient and caregiver focus groups and interviews, with overall themes described below. Themes from each forum are detailed in four appendices, including sampling of reflective patient and caregiver quotes. See Table 5 for a full list of patient and caregiver themes and selected quotes. See S3 Data.

**Theme 1. Distrust and lack of respect for medical professionals.** A broad distrust and lack of respect for medical professionals was the most commonly occurring theme across all focus groups and interviews. Many patients (n = 9) and caregivers (n = 7) detailed how this sentiment was catalyzed by long journeys to a proper diagnosis, including misdiagnosis, medical mistreatment that negatively affected course of NMOSD, false prognoses, minimization and undertreatment of pain symptoms, forced reliance on other patients and/or social media for disease education, and caregivers having to assume the role of patient advocate and/or

**Table 4. Paired patient-caregiver correlations.**

| Patient Scores | Caregiver Scores | | | | | | |
|---|---|---|---|---|---|---|---|
| | AAQ | CFQ | BAI | BDI | UC Coping | OC Coping | ACE |
| AAQ | .118 | -.014 | -.041 | .191 | -.153 | -.078 | -.068 |
| CFQ | .049 | -.077 | -.140 | .052 | -.062 | -.453^ | .304 |
| BAI | -.118 | .033 | .617** | .196 | .023 | -.090 | -.152 |
| BDI | -.031 | -.365 | .313 | .403^ | -.456* | -.187 | -.056 |
| UC Coping | .472* | -.142 | -.151 | .067 | .071 | -.149 | .310 |
| OC Coping | .014 | -.205 | -.353 | -.264 | .032 | .512* | -.107 |
| ACE | .451^ | .217 | -.140 | .392 | -.177 | .065 | .000 |

Abbreviations: AAQ, Acceptance and Action Questionnaire-II; CFQ, Cognitive Fusion Questionnaire-13; BAI, Beck Anxiety Inventory; BDI, Beck Depression Inventory-II; UC, Undercontrolled; OC, Overcontrolled; ACE, Adverse Childhood Experiences.

^ < .1,

* < .05,

** < .01

**Table 5. Patient and caregiver themes with selected quotes.**

| Theme | |
|---|---|
| 1. Distrust & lack of respect for medical professionals | • "The first time I got sick, I ended up. . . in the hospital for 5 weeks, 3 different hospitals, 'cause I was sick for an entire month, and everybody kept sending me home. They kept [saying] we don't see anything. . . and sending me home." (38 y.o. female patient).<br>• "I had doctors tell me it's all in my head when I was paralyzed. . . my respect for doctors has really gone down the drain. . ." (28 y.o. female patient).<br>• "'Oh it's a pinched nerve, lose ten pounds and you'll feel better'"(51 y.o. female patient). |
| 2. Lack of support and resources immediately after diagnosis. | "'[After I was diagnosed] I went through a whole year of grief. I couldn't read anything online about NMO. I couldn't do anything for about seven months after my diagnosis. . . I felt like I was standing on the edge of a cliff, ready to jump 24/7. My body felt like it was a rubber band stretched to the hundredth degree, and I was just waiting. . . waiting for another attack" (56 y.o. female patient).<br>• "My immediate question was, am I going to die from this? I was kind of in like survival mode. It was a fight-or-flight moment. What do I do?" (33 y.o. female patient).<br>• "Facebook is a huge form of support for information. . . and that's very helpful, especially if I'm at an ER and the doctors are not trained on NMO"(35 y.o. female patient). |
| 3. Impact of NMOSD on relationships. | • "While she was going to [live-in] blind school,.. all of the others had lost their spouses. . . the partners couldn't deal. It strengthened us in the fact that we did this together. I kept the home front down"(49 y.o. male caregiver).<br>• "Because I look normal, and . . .have a cane. . .people walk up to me and they'll go, 'Oh what did you do your leg?' I get so tired of having to say, '. . .I have NMO.' (52 y.o. female patient).<br>• "We have family that can't deal with it. . . we have cut those ties"(49 y.o. male caregiver). |
| 4. Deviation from valued living. | • "That was the saddest thing for me. . . that we couldn't go see the Christmas lights as a family anymore because she could no longer see"(74 y.o. female caregiver).<br>"I just turned 25. . .you think 25 is the prime of life, and that everything is perfect, and you're the hottest, . . . and everything changed. I went from being the kind of person who didn't even take Tylenol to taking like 18 pills a day. I couldn't see, I had trouble walking, and my entire life changed overnight. . . I'm so young, it just doesn't seem fair"(28 y.o. female patient).<br>"I started getting really bad anxiety because I couldn't see people. . . can't read facial cues, or. . . body language. So I got a bunch of anxiety from going blind. . . I couldn't really participate in as much things as I used to" (56 y.o. female patient). |
| 5. Internalizing feelings as a coping strategy. | • "I am not very expressive about [NMO] unless it really comes to a boiling point where I can't hide it. . .I try not to talk about it or make it into a bigger deal than what I think is appropriate" (39 y.o. male patient).<br>• "I don't talk about [her NMO]. I just don't talk about it." (45 y.o. male caregiver).<br>• "One of the doctors told me that I needed to prepare for the worst. I just kept my energy high, just to not let her know that I was worried. . ." (61 y. o. male caregiver). |

mediator in healthcare settings. Interestingly, all patients who spoke about pain as a primary symptom (n = 6) also reported that their treating providers minimized and undertreated their pain.

**Theme 2. Lack of support and resources immediately after diagnosis.** Many patients (n = 9) and caregivers (n = 7) reported significant emotional distress immediately following diagnosis of NMOSD, resulting from various factors including isolation as a person with a rare

disease and a near-total lack of information provided by medical professionals, resulting in a reliance on other patients (typically via social media) to learn more about the disease. Numerous patients (n = 6) reported trauma symptomology when discussing initial diagnosis of NMOSD, including feelings of being in "survival mode", hypervigilance for symptoms, sleep disturbance, confusion, and agitation. Patients also reported depressive symptoms during this initial time post-diagnosis, including grief, sadness, anhedonia, feelings of guilt, and suicidal ideation.

**Theme 3. The impact of NMOSD on relationships.**    Patients and caregivers reported significant impact of NMOSD on relationships with friends and loved ones. For some, outward signs of disease had the greatest impact, including mood-related side effects of steroid treatment. For others the "invisible" nature of NMOSD symptoms impacted relationships most, including family members not respecting their limitations as they "look fine". Some participants (n = 4 patients, n = 4 caregivers) shared the positive impact of NMOSD on their patient-caregiver dyad due to shared feelings of resilience and teamwork.

**Theme 4. Deviation from valued living.**    Identifying personal values and living in accordance with them is a strong predictor of psychological health [41]. Many patient (n = 9) and several caregiver (n = 4) narratives included experiences of NMOSD interfering with their ability to align their personal values with their daily actions. Particularly among younger patients, feelings of an "identity crisis" and difficulty finding meaning in life post-diagnosis were frequently reported experiences. Patients (n = 5) and caregivers (n = 5) reported grief regarding NMOSD symptoms prohibiting the family from participating in favorite activities or traditions. Some caregivers (n = 4) also reported patient's immunocompromised status impacted their experience of the COVID pandemic, with particular emphasis on being unable to see each other in person for extended periods of time.

**Theme 5. Internalizing feelings as a coping strategy.**    Internalizing or suppressing emotional experience is largely associated with reduced psychological health. Male patients and caregivers in particular, reported relying on internalizing feelings about NMOSD in order to cope with the burden of the disease. Several male patients (n = 2) and caregivers (n = 4) shared sentiments that sharing their feelings about NMOSD would be a negative experience, with caregivers reporting masking feelings of sadness or worry related to their loved one's NMOSD, to project a sign of strength to their loved ones.

## Discussion

Findings from focus groups, individual interviews, and standardized self-report measures, suggest a more complex and nuanced psychological burden among patients with NMOSD and their caregiver/loved ones than that described in empirical literature to date. Nearly three quarters of NMOSD patients and nearly half of their caregivers reported clinically relevant anxiety, which is a higher rate than expected based on literature [11]. The context and source of anxiety, as described by patients and caregivers, commonly centers around the unpredictable nature of subsequent NMOSD attacks, the progressive nature of the disease, and a lack of information about what to expect at the time of diagnosis.

A third of patients endorsed clinically-relevant depressive symptoms, which is similar to that reported in the literature to date. While specific prompts for depression are not reported in the literature to date, patients detailed their grief reactions upon diagnosis, depression regarding deviation from values-based living, and regarding stigma related to having a misunderstood and often "invisible" illness.

A previously unknown aspect of the psychological burden of NMOSD involves cognitive fusion, which can be understood as rigid attachment to thoughts as truth. For example, having

the thought "I can't do anything because of NMOSD" and allowing this thought to prevent oneself from attempting new activities. Most patients and caregivers endorsed clinically-relevant cognitive fusion, with associated high levels of anxiety, suggesting these could be important mental health treatment targets.

The prevalence of over-controlled coping among patients with NMOSD has not been examined to date. The majority of patients and all caregivers reported relying primarily on over-controlled coping styles. Over-controlled coping is associated with rigidity, continued perseverance with ineffective coping skills, internalizing emotions, and distress overtolerance [43]. Patients and caregivers reported persistent stoicism as a coping strategy, including suppressing intense emotions in the presence of their loved ones. Stoicism and overcontrol of emotions, is linked to help-seeking delays, inadequate pain treatment, and caregiver strain [43, 45]. Also, patient and caregiver anxiety and over-controlled coping were strongly associated, suggesting that if one member of the dyad is anxious or inflexible in their approach to managing NMOSD-related stressors, the other member is likely to respond similarly, rather than to provide a calming or flexible counter-perspective.

Another previously unknown aspect of the psychological burden of NMOSD elucidated by this study is the significant stress reaction reported by patients and caregivers immediately following diagnosis. Symptoms reported align with posttraumatic stress disorder (PTSD), including avoiding mention of NMOSD, somatic hypervigilance, and ongoing medical procedures acting as re-traumatization. One respondent experienced sudden onset suicidal ideation at this time. The perception of diagnosis and medical procedures as traumatic is particularly salient given that about a third of patients and caregivers endorsed a clinically significant history of adverse childhood events, including household members living with substance abuse or mental illness, verbal or emotional abuse, and parental divorce. These adverse experiences are associated with increased risk of chronic health conditions, including autoimmune diseases [46].

Factors contributing to this exacerbation of stress include a lack of adequate information about what to expect at the time of diagnosis, total lack of mental health resources at that time, with no follow-up by their care teams regarding emotional health and coping. Almost all patients and caregivers expressed mistrust of medical professionals, as a result of misdiagnosis, minimizing of reported symptoms including pain, iatrogenic treatment, and lack of information as described above. Mistrust toward health care professionals affects patients' health-related outcomes [47] and is linked to underuse of treatment [48].

Deviation from value-based living is another novel aspect of the psychological burden of NMOSD, reported by both patients and caregivers. Living in accordance with one's values is a core component of physical and psychological well-being [49], correlating with resilience [50] and medication adherence [51], and negatively correlating with depression [52]. Patients tended to report larger discrepancies between how important a certain valued domain is to them (for example "being an involved grandparent") and their ability to live according to this value. Both patients and caregivers reported significant stress and grief regarding deviation from living their values since diagnosis of NMOSD. This loss of value-based living was most often caused by physical disabilities, although for some patients, it was also caused by NMOSD-related depression, anxiety, or trauma. Several patients described experiencing an "identity crisis" upon diagnosis. Caregivers reported finding fulfillment in their caregiving role, while also lacking replacement activities and hobbies for those they gave up in order to serve in this role.

Interestingly, impact on social relationships both exacerbated and eased the psychological burden of NMOSD. Patients experiencing outward symptoms of NMOSD, such as vision loss or impaired mobility, reported relationships with friends and family were negatively impacted by the stigma of disability. Patients with "invisible" symptoms of NMOSD, such as pain or

nausea, reported their relationships suffered from a lack of understanding of the burden of disease. There is a large body of research detailing the increase in depression and suicide risk associated with diminished social interaction and loss of relationships [53]. Interestingly, most dyads reported their relationship strengthened because of the diagnosis in NMOSD, even as relationships with others were negatively impacted. These dyads reported that a shared sense of resilience and learning to work as a team contributed to a closer relationship.

Finally, most patients and nearly half of their caregivers had a history of engagement in psychotherapy. However, prior psychotherapy experiences were not effective in meeting addressing their needs. An overwhelming majority of patients and caregivers expressed interest in an NMOSD-specific telehealth-delivered talk therapy, specifically as a patient/caregiver dyad.

## Limitations

Recruitment via patient advocacy organizations, social media, and word of mouth may have unintentionally recruited participants who are more open to discussing mental health and/or personal experiences. In addition, proof of diagnosis of NMOSD was not required for participation, with self-reported diagnosis (typically accompanied by membership in an NMOSD-specific social media forum) sufficing. Also, despite efforts to include participants with diverse racial and ethnic identities, the majority of our sample was Caucasian, whereas NMOSD is more prevalent among nonwhite populations [6].

## Conclusions and future directions

Adjustment to and life with NMOSD poses a unique psychological burden, both to individuals living with this disease, and to their loved ones. Assessment and linkage with mental health services at the time of diagnosis is strongly recommended. Anxiety, depression, cognitive rigidity, over-controlled coping, departure from value-based living, and exacerbated stress reactions were detailed by our sample of people living with NMOSD and their caregivers. Additionally the interconnection between patient and caregiver coping and burden results in anxiety and rigid control strategies for the dyad.

Given the extensive and interconnected psychological burden of NMOSD experienced by patients and their loved ones, and the reported interest in a dyadic intervention, we propose a caregiver-assisted, NMOSD-specific mental health intervention, delivered via telehealth in order to accommodate physical limitations associated with NMOSD. Specifically, Acceptance and Commitment Therapy (ACT) [54], adapted to address the specific psychological burden of NMOSD as outlined in our findings, is one potential intervention. This novel, experiential talk-therapy has proven effective in the management of other illness states [55–58]. With an emphasis on values clarification, taking steps toward personal values despite illness-related obstacles, un-sticking from problematic thought content, and acceptance of what can not be changed, ACT, adapted for NMOSD, has high potential to reduce depression, stress, and caregiver strain, and improve treatment adherence and quality of life.

Additionally, we propose that immediate attention be devoted to repairing the distrust that many patients with NMOSD and their caregivers have toward the medical system, often due to misdiagnosis and resulting iatrogenic treatment. Given newer testing and diagnostic protocols, hopefully a new generation of patients with NMOSD will not endure such a "diagnostic odyssey". In the meantime, establishment of a vetted, centralized location for detailed patient education and resource materials, including linkage with effective mental health support is needed urgently, and should be shared at the time of diagnosis. In addition, further education of providers is required, regarding the specific psychological burden of NMOSD, including anxiety regarding subsequent attacks and progression, a tendency toward stoicism despite pain,

anxiety, and depression, and the need to support and involve the caregiver to prevent burnout and enhance the patient-caregiver support system.

## Supporting information

**S1 Data.**
(XLSX)

**S2 Data.**
(XLSX)

**S3 Data. Patient focus group qualitative data.**
(DOCX)

## Author Contributions

**Conceptualization:** Darcy C. Esiason, Ankita Deshpande, C. Virginia O'Hayer.

**Data curation:** Darcy C. Esiason, Nicole Ciesinski, Chelsi N. Nurse, C. Virginia O'Hayer.

**Formal analysis:** Darcy C. Esiason, Nicole Ciesinski, C. Virginia O'Hayer.

**Funding acquisition:** Darcy C. Esiason, Tom Hattrich, Ankita Deshpande, C. Virginia O'Hayer.

**Investigation:** Darcy C. Esiason, Chelsi N. Nurse, C. Virginia O'Hayer.

**Methodology:** Darcy C. Esiason, C. Virginia O'Hayer.

**Project administration:** Darcy C. Esiason, Nicole Ciesinski, Chelsi N. Nurse, Wendy Erler, C. Virginia O'Hayer.

**Supervision:** Darcy C. Esiason, C. Virginia O'Hayer.

**Validation:** Nicole Ciesinski.

**Writing – original draft:** Nicole Ciesinski, C. Virginia O'Hayer.

**Writing – review & editing:** Nicole Ciesinski, Chelsi N. Nurse, Wendy Erler, Tom Hattrich, Ankita Deshpande.

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
