## [Decision Letter · Decision Letter 0]

20 Sep 2023

PONE-D-23-06896The Psychological Burden of NMOSD - As Defined by Patients and CaregiversPLOS ONE

Dear Dr. o'hayer,

Thank you for submitting your manuscript to PLOS ONE. After careful consideration, we feel that it has merit but does not fully meet PLOS ONE’s publication criteria as it currently stands. Therefore, we invite you to submit a revised version of the manuscript that addresses the points raised during the review process.

We look forward to receiving your revised manuscript.

Kind regards,

Teresa Juárez-Cedillo, PhD

Academic Editor

PLOS ONE

Journal Requirements:

“This study generously was supported by a grant from Alexion AstraZeneca Rare Disease.”

“CVO'H, DCE, NKC, CNN received an award (awarded to CVO'H as the PI) from Alexion Pharmaceuticals to fund this study (alexion.com). While the funders had no role in study design, data collection & analysis, decision to publish, or preparation of the manuscript - they were given 30 days in which to review the manuscript prior to publication.”

Additional Editor Comments (if provided):

It is a relevant study since it considers the perspective of both patients and their caregivers, however, there are some points that must be considered before being able to make a decision.

The introduction should be focused on the psychological burden of NMOSD and the main findings found so far.

Authors must define whether they will only include qualitative data or also quantitative data in the writing.

If they only include quantitative data think that is the objective of the study).

Adequately describe the methodology, the population selection criteria, as well as the qualitative methodology technique used to obtain the sample size.

In the analysis section, clearly describe the type of analysis carried out on the data obtained (what theory was the analysis based on), as well as the software used for the analysis.

Choose the testimonies of the participants based on their relevance to respond to the objective of the study.

Carry out the discussion based on these findings identified in the testimonies in accordance with the theory chosen for their analysis.

Review references that comply with the requirements of the journal https://journals.plos.org/plosone/s/submission-guidelines

If, on the other hand, it is decided to use both quantitative and qualitative data, it is necessary to restructure the writing based on a mixed study and substantiate both the methodology and the data analysis.

Reviewers' comments:

Reviewer's Responses to Questions

**Comments to the Author**

1. Is the manuscript technically sound, and do the data support the conclusions?

Reviewer #1: Yes

Reviewer #2: Partly

2. Has the statistical analysis been performed appropriately and rigorously? 

Reviewer #1: Yes

Reviewer #2: No

3. Have the authors made all data underlying the findings in their manuscript fully available?

Reviewer #1: Yes

Reviewer #2: No

4. Is the manuscript presented in an intelligible fashion and written in standard English?

Reviewer #1: Yes

Reviewer #2: Yes

5. Review Comments to the Author

Reviewer #1: This paper describes the psychological burden of NMOSD in patients and caregivers. Alot of work has clearly gone into this. The manuscript is detailed with good insights and clear themes, and provides viewpoints from both patient and caregiver perspective (although having the physician perspective would have provided another source of triangulation).

In particular, I like how many critiques of the healthcare system were raised, and this honesty highlights also the trust participants had in the research team, to be able to freely share their comments.

Some major comments

The paper is excessively lengthy throughout, especially the introduction, results and discussion. Although there is no word limit imposed by the journal, such a length is cognitively draining. Most other qualitative articles published in PLOS ONE look to be about 2500-3500 words.

Secondly, the overall aims of the study are not clearly demonstrated – what is the clinical question and what are the gaps that the study is trying to address?

Other comments

In the abstract, it can be clearer that the 16 patients and 11 caregivers who did semi-structured interviews were a subset of the 31 adults/22 caregivers (if indeed so)

I feel that the introduction is too long. Given that the paper focuses on the psychological burden of NMOSD, I find that the section detailing the factors that contribute to the debilitating nature of NMOSD (page 4-7) may be unnecessary

Given that the psychological burden of NMOSD is fairly well established as authors alluded to, what is the purpose of the current study, e.g gaps in current literature and how the study aims to address these gaps can be better detailed.

The methodology could be more clearly fleshed out.

- Were the individual interviews concurrent or sequential to the focused group interviews?

- Was there any exclusion criteria?

Additionally, important elements of a rigorous qualitative study (memo writing, participant checking, iterative analysis, theoretical saturation, coreq checklist etc) appear to be missing

“All 30 patient participants completed a demographics survey. “

- 30 or 31?

Results started off with quantitative analyses of 30 patient survey responses, followed by 21 caregiver surveys. I personally dont think this should be the main focus of the manuscript as the sample is quite small. Furthermore, many of the statistical analyses appeared to have been conducted without a clear hypothesis.

I’m quite confused with the analysis of qualitative data. The way it is written on page 22 seems to suggests that qualitative thematic analysis was only conducted for those who did the individual interviews (16 patients and 11 caregivers)

The results would be better represented as a table of themes with select highlighted quotes (with a full table as a supplementary file). Within the manuscript text itself, writing should be more concise and quotes can be shortened. - Instead of quoting full quotes from patients, a better way would be to pick out portions of the sentence which they specifically demonstrate the theme – for e.g. page 21 under point 4.1 – the first quote could go without the portion regarding the nurses.

Alot of the discussion section reiterates and repeats findings from the results. It may be better to discuss instead what is new and how the results build on existing literature.

Furthermore, given that there were so much negative comments toward healthcare providers, how do authors think the healthcare system could be improved to provide a better experience for NMOSD patients?

I do like the section on future directions which provide practical suggestions for implementation, although like the rest of the manuscript, this could be more concisely expressed.

Reviewer #2: EVALUATION

Thank you for the opportunity to review this paper exploring the Psychological Burden of NMOSD - As Defined by Patients and Caregivers. The study makes some unique contributions to literature, although I have major concerns with the paper in its current format that will need to be addressed prior to being considered for publication. The primary concerns are the following:

1. The manuscript did not follow submission guidelines as detailed on the Plos One website (https://journals.plos.org/plosone/s/submission-guidelines). It should be organized following Plos One formatting and lines of the manuscript should be also numbered to make review easy.

2. The authors collected qualitative and quantitative data, and this means that they employed a mixed methods approach. However, in the methods section, they are not clear about an employed research design, the sampling strategies for both quantitative and qualitative data, and the procedures of data collection, analysis, and discussion. You can refer to the suggested book from 203 pages to improve your manuscript.

TITLE

3. The study mixed qualitative and quantitative data. However, this is not quickly identified by the title.

ABSTRACT

4. The abstract does not make it clear what the rationale of the study is and why collecting both qualitative data and quantitative data. What did the qualitative study provide that the quantitative study did not and vice-versa? How does it add value?

5. “This study aims to describe and understand the psychological burden of patients with NMOSD and their caregiver/loved ones via self-report measures, focus groups, and in-depth interviews”. I propose: This study aims to investigate and understand the psychological burden of patients with NMOSD and their caregivers/loved ones from…………

6. Research design, population, sampling techniques, sample, study instruments, estimation techniques, and software for data storage and analysis should also be highlighted in the abstract.

INTRODUCTION

7. The authors should describe variables in chronological and logical sequences rather than creating their subheadings with a lack between them.

8. Also, the authors should be clear on the rationale for the study. It is not enough to say “empirical literature and patient media perspectives suggest a significant psychological burden associated with diagnosis of NMOSD. There is a clear need to develop a comprehensive understanding of this experience, both from patient and caregiver perspectives.” It would be helpful if the introduction reviewed some of the existing literature on the Psychological Burden of NMOSD among Patients and their Caregivers by highlighting at the same time the existing gap that led to collecting qualitative and quantitative data.

METHODS

9. In the participants' section, please incorporate the research design, population, sampling techniques, sample (also summary statistics describing the sample), inclusion, and exclusion criteria where applicable.

10. Also, include statistics and software for data storage and analysis in the statistical analysis part.

11. It is not clear why using individual interviews and focus groups to collect qualitative data

12. The authors stated, “Both patient and caregiver participants were compensated via $150 gift card for completion of self-report measures and participation in an individual interview.” However, they should provide ethically accepted reasons for providing this amount and the reference supporting their reason.

13. Authors should report the value of Cronbach’s alpha of the study instruments as well as their cutoff scores to easily classify the participants into those with significant symptoms of psychological burden and those without significant symptoms.

RESULTS

14. I think tables can help present clearly and summarise the authors’ findings. Why didn’t you use the tables?

DISCUSSION

15. The referencing style should be Elsevier Vancouver

6. PLOS authors have the option to publish the peer review history of their article (what does this mean?). If published, this will include your full peer review and any attached files.

Reviewer #1: **Yes: **Ellie Choi

Reviewer #2: No

---

## [Decision Letter · Decision Letter 1]

27 Feb 2024

PONE-D-23-06896R1The psychological burden of NMOSD – a mixed method study of patients and caregiversPLOS ONE

Dear Dr. o'hayer,

Thank you for submitting your manuscript to PLOS ONE. After careful consideration, we feel that it has merit but does not fully meet PLOS ONE’s publication criteria as it currently stands. Therefore, we invite you to submit a revised version of the manuscript that addresses the points raised during the review process.

**ACADEMIC EDITOR: **

Minor revisions are required. Reviewing the length of the introduction focusing on considering the most relevant elements reported in the literature.

All data analyzed from the study population must be in the results section.

Combine the conclusion with the discussion and write a new conclusion with the most relevant findings of this study.

The authors must consider the observations made by the reviewers.

We look forward to receiving your revised manuscript.

Kind regards,

Teresa Juárez-Cedillo, PhD

Academic Editor

PLOS ONE

Journal Requirements:

Reviewers' comments:

Reviewer's Responses to Questions

**Comments to the Author**

1. If the authors have adequately addressed your comments raised in a previous round of review and you feel that this manuscript is now acceptable for publication, you may indicate that here to bypass the “Comments to the Author” section, enter your conflict of interest statement in the “Confidential to Editor” section, and submit your "Accept" recommendation.

Reviewer #1: (No Response)

Reviewer #2: All comments have been addressed

2. Is the manuscript technically sound, and do the data support the conclusions?

Reviewer #1: Yes

Reviewer #2: Yes

3. Has the statistical analysis been performed appropriately and rigorously? 

Reviewer #1: Yes

Reviewer #2: Yes

4. Have the authors made all data underlying the findings in their manuscript fully available?

Reviewer #1: Yes

Reviewer #2: No

5. Is the manuscript presented in an intelligible fashion and written in standard English?

Reviewer #1: Yes

Reviewer #2: No

6. Review Comments to the Author

Reviewer #1: Unfortunately I cant seem to access the author's reply to the previous comments.

For the current manuscript, the front component/introduction is quite lengthy and difficult to get through. I'm not sure there is a need to go through paragraphs of each individual construct and the existing literature (e.g. coping, psychological resilience). I feel that it is more important to synthesise the literature to highlight the gaps in knowledge.

I'm not sure what happened to the tables/figures, have they been removed from the previous version?

Reviewer #2: Thank you for allowing me to read this revised manuscript. Having reread the manuscript, the authors have addressed most of my comments. However, there are a few comments to address before heading to another step.

7. PLOS authors have the option to publish the peer review history of their article (what does this mean?). If published, this will include your full peer review and any attached files.

Reviewer #1: No

Reviewer #2: No

---

## [Author Response · Author response to Decision Letter 1]

4 Mar 2024

Dear Dr. Juárez-Cedillo,

Thank you for the detailed feedback regarding our manuscript, “The psychological burden of NMOSD – a mixed method study of patients and caregivers” (PONE-D-23-06896R1). We are aware of the considerable amount of time and effort that go into reviewing a manuscript, and we are very appreciative of the attention to our work and the thoughtful suggestions made. 

Below are our responses to the reviewer comments. We are confident that a more concise and focused manuscript results from the reviewer’s thoughtful feedback. 

ACADEMIC EDITOR:

COMMENT 1: Reviewing the length of the introduction focusing on considering the most relevant elements reported in the literature.

RESPONSE TO COMMENT #1: Thank you – we have condensed the introduction considerably, focusing only on 

COMMENT 2: All data analyzed from the study population must be in the results section.

RESPONSE TO COMMENT #2: Yes – these are now all included in the results section. 

COMMENT 3: Combine the conclusion with the discussion and write a new conclusion with the most relevant findings of this study.

RESPONSE TO COMMENT #3: Thank you – we have created a new Conclusions and Future Directions section to address this feedback. 

Reviewer #1: 

COMMENT 1: The front component/introduction is quite lengthy and difficult to get through. I'm not sure there is a need to go through paragraphs of each individual construct and the existing literature (e.g. coping, psychological resilience). I feel that it is more important to synthesise the literature to highlight the gaps in knowledge.

RESPONSE TO COMMENT #1: Agree – we have condensed the intro considerably. 

Reviewer #2: 

COMMENT 1: Page 4-5: Generally, paragraphs are five to ten lines in length. A paragraph of two to four lines doesn’t look smart.

RESPONSE TO COMMENT #1: Agree – we have consolidated paragraphs in the newly revised and condensed introduction. 

COMMENT 2: Pages 7-8: Socio-demographic characteristics of the participants should be removed from the materials and methods section to be pasted and analyzed in the results section.

RESPONSE TO COMMENT #2: Thank you for this correction, we have moved this information and associated Tables 1 and 2 accordingly, to the results section. 

COMMENT 3: Pages 14-18: Information regarding descriptive analysis of study variables and the rate of psychological burden was clearly mentioned in the manuscript. However, there are no tables summarizing this information.

RESPONSE TO COMMENT #3: Thank you – we have created such a table summarizing the descriptive analysis of study variables. This is now Table 3. 

COMMENT 4: Also, can you provide information on other measurements of variability such as standard deviation, skewness, and kurtosis? Descriptive analysis of study variables based only on the mean doesn’t provide much information.

RESPONSE TO COMMENT #4: Yes, agree - we have included these measures of variability (standard deviation, skewness and kurtosis) in our new Table 3. 

COMMENT 5: Pages 22-25: It seems that your conclusion section reflects the discussion section. The reason is that you have explained what you found, showing how it correlates with the existing literature, which makes it a discussion section. Consequently, you can then set up a conclusion section apart to summarize your findings and explain why they are very important. Then, I suggest the following sections: Results, Discussion, Conclusions,…

RESPONSE TO COMMENT #5: Thank you for this suggestion. We have now differentiated Results from Discussion sections and, rather than creating much additional text, we have created an integrated Conclusions and Future Directions section.

We have electronically submitted the revised manuscript. We hope that we have addressed all comments and suggestions appropriately. If not, we will be happy to consider making whatever additional changes you or the reviewers request. Thank you for continuing to consider this manuscript for publication in PLOS ONE. We look forward to hearing from you.

Sincerely,

C. Virginia O’Hayer, Ph.D.

Clinical Professor

Director, Jefferson Center City Clinic for Behavioral Medicine

Department of Psychiatry & Human Behavior

Thomas Jefferson University Hospital

---

## [Editor Report · Decision Letter 2]

6 Mar 2024

The psychological burden of NMOSD – a mixed method study of patients and caregivers

PONE-D-23-06896R2

Dear Dr. o'hayer,

We’re pleased to inform you that your manuscript has been judged scientifically suitable for publication and will be formally accepted for publication once it meets all outstanding technical requirements.

Kind regards,

Teresa Juárez-Cedillo, PhD

Academic Editor

PLOS ONE
---

## [Editor Report · Acceptance letter]

19 Mar 2024

PONE-D-23-06896R2 

PLOS ONE

Dear Dr. o'hayer, 

I'm pleased to inform you that your manuscript has been deemed suitable for publication in PLOS ONE. Congratulations! Your manuscript is now being handed over to our production team.

Kind regards, 

on behalf of

Dr. Teresa Juárez-Cedillo 

Academic Editor

PLOS ONE